

# Specific lifestyle factors and in vitro fertilization outcomes in Romanian women: a pilot study

Iulia A. Neamtiu[1,2,*], Mihai Surcel[3,*], Thoin F. Begum[4],
Eugen S. Gurzau[1,5], Ioana Berindan-Neagoe[5], Cornelia Braicu[5],
Ioana Rotar[3], Daniel Muresan[3] and Michael S. Bloom[6]

[1] Health Department, Environmental Health Center, Cluj-Napoca, Romania
[2] Faculty of Environmental Science and Engineering, Babes-Bolyai University, Cluj-Napoca, Romania
[3] 1st Obstetrics and Gynaecology Department, Iuliu Hatieganu University of Medicine and Pharmacy, Cluj-Napoca, Romania
[4] Department of Environmental Health Sciences, University at Albany, State University of New York, Rensselaer, New York, United States
[5] Research Center for Functional Genomics, Biomedicine and Translational Medicine, Iuliu Hatieganu University of Medicine and Pharmacy, Cluj-Napoca, Romania
[6] Department of Global and Community Health, George Mason University, Fairfax, Virginia, United States
* These authors contributed equally to this work.

Corresponding author
Iulia A. Neamtiu,
iulianeamtu@ehc.ro

## ABSTRACT

**Background:** Infertility is an important health concern worldwide. Although lifestyle habits and behaviors have been widely reported as predictors of IVF outcomes by previous studies, they have not been reported for Romanian women undergoing IVF. In this regard, our pilot study aimed to begin to address the data gap by assessing lifestyle predictors of *in vitro* fertilization (IVF) outcomes in Romanian women.
**Study design:** Our pilot study included 35 participants who completed a first IVF cycle at a single infertility center. We evaluated individual self-reported lifestyle habits and behaviors as predictors of IVF outcomes, and employed principal component analysis (PCA) to characterize multiple lifestyle habits and behaviors into personal care product (PCP) use, and healthy diet and physical activity patterns as predictors of IVF outcomes.
**Results:** Our PCA analysis showed that greater use of PCPs was associated with lower probabilities of pregnancy (RR: 0.92, 95% CI [0.87–0.98]) and live birth (RR: 0.94, 95% CI [0.88–1.01]) while, the healthy dietary habits and physical activity were associated with a higher likelihood of pregnancy, although without statistical significance (RR: 1.10, 95% CI [0.93–1.30]).
**Conclusions:** In this pilot study we identified associations between IVF outcomes among Romanian women and certain lifestyle habits and behaviors including stress, diet and physical activity, and certain PCP use. We also estimated the joint effects of multiple lifestyle factors using PCA and found that PCP use, healthy dietary habits and physical activity were associated with IVF outcomes.

## INTRODUCTION

Many couples worldwide face impaired fecundity and infertility problems. Studies of global prevalence estimate that infertility affects 186 million people (*Inhorn & Patrizio, 2015*), between 8% and 12% of reproductive-aged couples worldwide (*Ombelet et al., 2008*), and 20–30% of reproductive-aged women in modern society (*United Nations D of E & SAPD, 2015*). Clinical infertility is defined by the World Health Organization as the failure to conceive a clinically recognized pregnancy after 12 months of regular unprotected heterosexual intercourse (*Zegers-Hochschild et al., 2009*). Due to the difficulties experienced in getting pregnant, women, including in Romania, increasingly resort to assisted reproductive technologies (ART), primarily *in vitro* fertilization (IVF), to achieve a pregnancy (*The European IVF-Monitoring Consortium (EIM) for the European Society of Human Reproduction and Embryology (ESHRE) et al., 2021*).

In all European countries, Medically Assisted Reproduction is governed by legislation, either independently or as part of a wider legislative framework. Most countries also have professional guidelines (*European Society of Human Reproduction & Embryology (ESHRE), 2017*). In Romania, as in other EU countries, infertility treatments (including IVF) are conducted according to national best practice guidelines (*Romanian Society of Obstetrics & Gynecology RC of P, 2019*), which are based on ESHRE recommendations (*Bosch et al., 2020*). In particular, Romanian, specialists tend to prioritize stimulation protocols that maximize the number of oocytes retrieved, with active management to ensure patient safety. GnRh antagonist protocols are preferred for pituitary control and embryos may be frozen for transfer in a subsequent cycle when the risk of ovarian hyperstimulation is high (*Romanian Society of Obstetrics & Gynecology RC of P, 2019*). As in other EU countries, couples beginning an IVF protocol in Romania are given a set of lifestyle recommendations based on ESHRE recommendations, such as: quitting cigarette smoking, losing weight, limiting alcohol consumption and sometimes, changing jobs that involve exposure to reproductive toxicants. Still, due to its short duration, IVF, unlike other types of infertility treatment, often occurs over an insufficient period of time to allow for meaningful changes in lifestyle.

Lifestyle factors such as diet, physical activity, cigarette smoking, personal care product (PCP) use, and exposure to environmental contaminants have been associated with women's fertility and IVF outcomes (*Domar et al., 2012*; *Firns et al., 2015*; *Homan, Davies & Norman, 2007*; *Hornstein, 2016*). Unbalanced and hypercaloric diets combined with a decline in the level of physical activity are important contributors to the increased frequencies of overweight and obesity among women worldwide. Obesity-related changes in insulin, leptin, and sex steroid hormone balances may impair ovarian function, follicular growth, and embryo implantation and thereby impact IVF (*Pasquali, 2003*; *American College of Obstetricians & Gynecologists (ACOG), 2005*; *Pasquali & Gambineri, 2006*). Reproductive toxicants in tobacco smoke may affect endometrial vascularization and myometrial relaxation, resulting in IVF implantation failure and pregnancy loss (*Dechanet et al., 2011*). PCPs, including lotions, fragrances, and cosmetics, contain chemical agents such as phthalates, parabens, bisphenols, toxic trace elements, and ultraviolet filters

(*Juhász & Marmur, 2014*; *Begum et al., 2020*), which have been associated with altered reproductive function (*Barrett, 2005*; *Borowska & Brzóska, 2015*; *Gore et al., 2015*) and IVF outcomes (*Bloom et al., 2010*, *2012a*, *2012b*; *Ehrlich et al., 2012*; *Hauser et al., 2016*; *Begum et al., 2020*; *Butts et al., 2021*). A growing body of evidence suggests that exposure to the chemical agents found in various consumer products may affect reproductive health, though the overall contribution of those agents to infertility is unknown (*di Renzo et al., 2015*).

Although lifestyle habits and behaviors have been widely reported as predictors of IVF outcomes by previous studies (*Homan, Davies & Norman, 2007*; *Hornstein, 2016*), they have not been reported for Romanian women undergoing IVF. This pilot study aimed to begin to address the data gap by assessing lifestyle predictors of IVF outcomes in Romanian women. Our study results may help direct a more comprehensive and definitive future investigation, with the ultimate goal of improving live birth rates among Romanian IVF patients.

## MATERIALS AND METHODS

Our pilot study included 35 participants who completed a first IVF cycle at a single infertility center. All study participants completed a physician-administered questionnaire. We evaluated individual questionnaire self-reported lifestyle habits and behaviors as predictors of IVF outcomes, and employed principal component analysis (PCA) to characterize multiple lifestyle habits and behaviors into personal care product (PCP) use, and healthy diet and physical activity patterns as predictors of IVF outcomes.

### Study sample, questionnaire, and clinical protocol

We enrolled 35 women who completed a first IVF cycle within the Assisted Reproduction Department at the 1st Obstetrics and Gynecology Clinic in Cluj-Napoca, Romania, between April 8th and October 1st, 2019. All study participants provided written informed consent prior to participation in our study, and the research protocol was approved by the "Iuliu Hatieganu" University of Medicine and Pharmacy Ethics Committee (Institutional Review Board (IRB) approval number 51, March 11th, 2019). We approached 37 women in total and had two participation refusals (94.6% participation rate). All study participants completed a physician-administered questionnaire. The questionnaire included 47 questions organized in six sections: demographic data (*e.g.*, name, address, and date of birth); socioeconomic data (*e.g.*, last graduated school); home and employment data (*e.g.*, questions on potential exposure to xenobiotics at home and in the workplace); health status (*e.g.*, medical and reproductive histories); lifestyle factors (*e.g.*, smoking, physical activity, use of PCPs, and dietary habits) and questions about the male partner (*e.g.*, age and medical history).

We abstracted clinical data from the participant medical file on antral follicle count (AFC) as an indicator of ovarian reserve, levels of hormones (follicle stimulating hormone (FSH), luteinizing hormone (LH), anti-Mullerian hormone (AMH), and estradiol), response to ovarian stimulation (peak estradiol, number of oocytes retrieved, and thickness

of the endometrial mucosa), and IVF endpoints (oocytes fertilized, number and quality of embryos, chemical pregnancy, clinical pregnancy, and live births).

Urine, blood, ovarian follicular fluid, and endometrial flushing fluid specimens were collected at the time of oocyte retrieval, for future metal and genetic analysis. All biospecimens were immediately processed after collection (centrifugation for blood serum separation) and stored at either −20 °C for future analysis of metals, or −80 °C for future genetic analyses.

Patients underwent controlled ovarian stimulation following a baseline infertility evaluation. They were assigned to a clinical protocol according to individual phenotype and infertility characteristics, reproductive history, and patient preference. Either a long protocol (ovarian down regulation with agonist gonadotropin-releasing hormone (GnRH)) or antagonist GnRH protocol was implemented. The starting gonadotropin doses (Menopur (Menotropin) or Follitropin-α (Gonal-F)) were tailored for patient age, AMH level, number of small antral follicles, body mass index (BMI), and reproductive history. Ovarian stimulation was monitored by transvaginal ultrasound and serum estradiol levels, beginning 4/5 days after initiation and the gonadotropin doses were adjusted according to the ovarian response. For final oocyte maturation, hCG (choriogonadotropin-α (Ovidrel)) was administered when more than three ovarian follicles exceeding 17 mm in diameter were detected by transvaginal ultrasound. Oocytes were retrieved by transvaginal follicle puncture 34–38 h after hCG administration. Collected oocytes in metaphase II arrest were fertilized by intracytoplasmic sperm injection (ICSI), or conventional insemination using fresh sperm from the male partner, 4–6 h after retrieval. Couples with male pathology (<2 million motile spermatozoa) or (≤4% normal morphology) received ICSI. Oocytes were evaluated 14–16 h after injection, and were graded as either normally fertilized (two pronuclei) or abnormally fertilized (single or poly pronuclear). Resulting embryo quality was assessed using the Istanbul consensus scoring system: I (best), II (good), III–IV (poor), and V (worst) (*Balaban et al., 2011*; *American Society for Reproductive Medicine (ASRM) Practice Committee & Society for Assisted Reproductive Technology Practice Committee, 2013*). Three or five days after the oocyte retrieval, one to two embryos were transferred to the patient's uterus, considering clinical factors and patient preference. The remaining embryos were frozen for future use. Lutinus 100 mg was administrated three times a day intravaginally, as support for the luteal phase. A pregnancy was biochemically confirmed by serum beta hCG measurement (beta hCG > 20 mIU/ml) 14 days after the oocyte retrieval. An ultrasound examination confirmed a "clinical pregnancy" 2 weeks later, by visualization of one or more gestational sacs (*Zegers-Hochschild et al., 2009*). Patients' obstetricians monitored the pregnancy and reported either spontaneous abortion or live birth.

## Statistical analysis

We calculated the fertilization proportion per woman (%) as the ratio between the number of fertilized oocytes and the number of retrieved oocytes, and the proportion of high-quality embryos per woman (%), as the ratio between the number of embryos (grades I and II) and the number of retrieved oocytes.

We first examined associations between 17 individual lifestyle factors and IVF outcomes among oocytes ($n$ = 194) and embryos ($n$ = 79). We operationalized responses to a physician-administered questionnaire as ordinal variables (*e.g.*, 1 for never to 5 for every day) to describe the daily to monthly frequencies of 17 lifestyle habits and the intensity of behaviors, including exposure to tobacco smoke and psychosocial stress, physical activity, use of PCPs, and dietary exposures. We then used principal component analysis (PCA) to describe the variability in patterns of women's lifestyle habits and behaviors. Based on their covariances, PCA summarizes a larger number of variables as a smaller number of independent "summary variables", factors that represent latent constructs (*Yong & Pearce, 2013*). We employed a varimax rotated PCA, based on a polychoric correlation matrix, to summarize 17 individual variables describing various lifestyle habits and behaviors (*Grace-Martin, 2019*). We selected two PCA factors summarizing women's use of PCPs, dietary habits and physical activity based on a scree plot, eigenvalues >2.0, and >10% variance explained by each factor (*DiStefano, Zhu & Mîndrilă, 2009*; *Yong & Pearce, 2013*). We only retained lifestyle habits and behaviors factor loadings >|0.5| in each PCA factor to ensure moderate-strong correlations to the summary PCA factors. We then multiplied each participant's value by the retained factor loading to calculate weighted sum scores for each summary PCA factor.

We used the two retained PCA factors as simultaneous predictors of IVF outcomes in multivariable regression models to estimate associations of lifestyle habits and behavior patterns with IVF outcomes, adjusted for age in years (*Polanska et al., 2014*), BMI in kg/m$^2$ (*Sarais et al., 2016*), current smoking status (*Domar et al., 2012*), and education as a measure of socioeconomic status (*Swift & Liu, 2014*), which were selected *a priori* as confounders based on the literature. We used Poisson regression with robust error variance employing generalized estimating equations (GEE) to estimate the relative risks (RR) and 95% confidence intervals (CI) for dichotomized IVF outcomes including oocyte fertilization, embryo quality, clinical pregnancy, and live birth as outcomes in relation to women's lifestyle habits and behavior patterns. The GEE accounted for "clustered" outcomes within each woman (*Zou, 2004*). Using negative binomial regression, we also calculated effect estimates for associations between baseline AFC and women's lifestyle habits and behavior patterns. We used linear regression to estimate the association between continuous baseline AMH levels (ng/mL), endometrial thickness (mm), and peak estradiol levels (pg/mL) as intermediate IVF outcomes with women's lifestyle habits and behavior patterns. We checked for and excluded influential observations defined as Dfbeta >|1.96|. We set statistical significance as $\alpha$ = 0.05 for a 2-tailed test. Statistical analysis was performed using SAS version 9.4 (SAS Institute Inc., Cary, NC, USA).

## RESULTS

Table 1 shows the distribution of demographic, and clinical factors among study participants. The women were between 27 and 44 years of age (mean and standard deviation (SD): 34.9 ± 4.9), while their male partners were somewhat older (38.1 ± 5.7 years). Most women ($n$ = 29) graduated university. There were 14 (40%) overweight or obese women among our study participants, and most were diagnosed with female factor

**Table 1 Distribution of demographic, and clinical factors among _n_ = 35 women undergoing IVF.**

|  | _n_ | _Mean ± SD (%)_ | _Min_ | _25th %_ | _50th %_ | _75th %_ | _Max_ |
|---|---|---|---|---|---|---|---|
| _Female age (years)_ | 35 | 34.9 ± 4.9 | 27.0 | 30.0 | 35.0 | 39.0 | 44.0 |
| _Male partner's age (years)_ | 35 | 38.1 ± 5.7 | 27.0 | 33.0 | 38.0 | 43.0 | 53.0 |
| **_Female BMI (kg/m² )_** |  |  |  |  |  |  |  |
| Underweight & Normal (<25) | 21 | (60.0) |  |  |  |  |  |
| Overweight & Obese (≥25)[a] | 14 | (40.0) |  |  |  |  |  |
| **_Education_** |  |  |  |  |  |  |  |
| Medium level of education[b] | 6 | (17.1) |  |  |  |  |  |
| High level of education[c] | 29 | (82.9) |  |  |  |  |  |
| **_Infertility Diagnosis_** |  |  |  |  |  |  |  |
| Female factor (including idiopathic) | 28 | (80.0) |  |  |  |  |  |
| Male factor | 3 | (8.6) |  |  |  |  |  |
| Mixed[d] | 4 | (11.4) |  |  |  |  |  |
| **_IVF endpoints_** |  |  |  |  |  |  |  |
| _Number of oocytes retrieved_ | 35 | 5.5 ± 3.7 | 0.0 | 2.0 | 5.0 | 8.0 | 16.0 |
| _Proportion of fertilized oocyte (%)_ | 35 | 74.3 ± 21.7 | 37.5 | 60 | 73.2 | 100 | 100 |
| _Number of embryos (grade I and II)[e]_ | 35 | 2.2 ± 1.5 | 0 | 1 | 2 | 3 | 7 |
| _Clinical Pregnancy_ | 10 | (28.6) |  |  |  |  |  |
| _Live birth_ | 6 | (17.1) |  |  |  |  |  |
| Peak estradiol (pg/mL) | 35 | 3,370.5 ± 1,196.2 | 1,560 | 2,350 | 3,210 | 4,200 | 7,830 |
| AMH (ng/mL) | 35 | 2.5 ± 1.6 | 0.3 | 1.3 | 1.9 | 3.6 | 5.9 |
| AFC (follicles) | 35 | 10.9 ± 4.9 | 4.0 | 8.0 | 11.0 | 12.0 | 25.0 |
| Thickness of endometrial mucosa (mm) | 35 | 9.1 ± 1.4 | 7.0 | 7.9 | 8.8 | 10.0 | 12.0 |

**Notes:**
AMH, anti-Mullerian hormone; AFC, antral follicles count; BMI, body mass index; PCOS, polycystic ovary syndrome; SD, Standard deviation.
[a] All obese women had BMI < 35.0 (Class I obesity).
[b] Medium level of education includes secondary school (_n_ = 1), vocational school (_n_ = 1), high school (_n_ = 1), and post high school (_n_ = 1).
[c] High level of education includes college (_n_ = 1) and faculty (_n_ = 1).
[d] PCOS with secondary male factor diagnosis (_n_ = 2), PCOS with secondary tubal factor diagnosis (_n_ = 1), and endometriosis with secondary male factor diagnosis (_n_ = 1).
[e] Grade I (best), grade II (good) based on the Istanbul consensus scoring system (_Balaban et al., 2011_).

infertility, including "idiopathic" (80%). A median of five oocytes (range 0–16) were collected from each woman, 74% of which were fertilized on average. A median of two grade I and II embryos were generated per couple. Ten (33.3%) women had a pregnancy, of whom six experienced a live birth (17.1%).

Table 2 shows the distribution of lifestyle habits and behaviors. Only a few women were current smokers (_n_ = 7), but 17 (48.6%) reported passive smoke exposure. Most of the women never smoked. Nineteen women reported experiencing medium (54.3%) daily stress and 14 women reported having high stress daily (40.0%). Almost half (45.7%) of the women did not exercise routinely. Of the 19 (54.3%) women who reported exercising weekly, most (74.3%) spend less than an hour during each exercise session. Half of the women used face cream daily (48.6%), followed by daily use of perfume (42.9%), body lotion (40.0%), and cleansing lotion (37.1%). The use of lip and eyeliner was infrequent in this population group (54.3 never using). Many women reported never using a foundation

**Table 2 Distribution of lifestyle habits and behaviors among n = 35 women undergoing IVF.**

| | n (%) |
|---|---|
| Female smoking status | |
| Current smoker | 7 (20.0) |
| Never smoker | 19 (54.3) |
| Former smoker | 9 (25.7) |
| Passive Smoking | 17 (48.6) |

| | Low | Medium | High | | |
|---|---|---|---|---|---|
| Level of stress | 2 (5.71) | 19 (54.3) | 14 (40.0) | | |
| | Never | <once/week | 1–2 times/week | 3–4 times/week | Everyday |
| Weekly frequency of exercising | 16 (45.7) | 2 (5.71) | 5 (14.3) | 6 (17.1) | 6 (17.1) |
| | <1 h | 1 h | 2 h | | |
| Duration of each workout | 26 (74.3) | 7 (20.0) | 2 (5.71) | | |
| | Never | <once/week | 1–2 times/week | 5–6 times/week | Everyday |
| Weekly use of face cream | 6 (17.1) | 3 (8.57) | 1 (2.86) | 8 (22.9) | 17 (48.6) |
| Weekly use of cleansing lotion | 12 (34.3) | 2 (5.71) | 2 (5.71) | 6 (17.1) | 13 (37.1) |
| Weekly use of body lotion | 9 (25.7) | 2 (5.71) | 2 (5.71) | 8 (22.9) | 14 (40.0) |
| Weekly use of perfume | 6 (17.1) | 4 (11.4) | 3 (8.57) | 7 (20.0) | 15 (42.9) |
| Weekly use of foundation cream | 20 (57.1) | 2 (5.71) | 2 (5.71) | 4 (11.4) | 7 (20.0) |
| Weekly use of lip and eyeliner | 19 (54.3) | 3 (8.57) | 1 (2.86) | 4 (11.4) | 8 (22.9) |
| Weekly use of mascara | 10 (28.6) | 3 (8.57) | 2 (5.71) | 7 (20.0) | 13 (37.1) |
| Weekly use of lipstick | 10 (28.6) | 5 (14.3) | 3 (8.57) | 3 (8.57) | 14 (40.0) |
| | <once/month | 2 times/month | Once/week | 3–4 times/week | Everyday |
| Monthly consumption of canned foods & beverages | 19 (54.3) | 8 (22.9) | 7 (20.0) | 1 (2.86) | – |
| Monthly consumption of fish | 17 (48.6) | 7 (20.0) | 10 (28.6) | 1 (2.86) | – |
| | <once/week | Once/week | 3–4 times/week | Everyday | |
| Weekly consumption of vegetables | – | 0 (0.0) | 14 (40.0) | 21 (60.0) | |
| Weekly consumption of fruits | 2 (5.71) | 1 (2.86) | 11 (31.4) | 21 (60.0) | |

cream (57.1%), but 37.1% did report using mascara and 40% used lipstick daily. Most women consumed fish less than once per month (48.6%). Most study participants consumed vegetables (60%) and fruits (60%) daily.

Table 3 describes the confounder-adjusted associations between individual lifestyle habits and behaviors and clinical IVF outcomes. We detected statistically significant associations for most lifestyle habits and behaviors as predictors of IVF outcomes, including pregnancy and live birth. Unexpectedly, past and passive tobacco smoke exposure were positively associated with pregnancy and live birth. Greater reported stress was associated with lower likelihoods of clinical pregnancy (RR: 0.29, 95% CI [0.18–0.46]) and live birth (RR: 0.21, 95% CI [0.11–0.43]). Greater consumption of fruits was associated with higher probabilities of pregnancy and live birth (RR: 1.50, 95% CI [1.01–2.24] and RR: 1.86, 95% CI [1.10–3.15], respectively). The unadjusted associations can be found in Table S1.

**Table 3 Associations between individual lifestyle habits and behaviors and clinical outcomes among _n_ = 35 women undergoing IVF, adjusted for current cigarette smoking status (yes/no), BMI, age, and highest level of education attained.**

Relative risk (95% CI)

| Outcomes | Years spent smoking in the past | Years of exposure to passive smoke | Stress level[a] | Routine weekly exercise | Workout duration (hours/per episode) | Monthly canned food/beverage consumption | Monthly fish consumption | Weekly vegetable consumption | Weekly fruit consumption |
|---|---|---|---|---|---|---|---|---|---|
| Oocyte fertilization | 1.02 [0.98–1.06] p-value = 0.37 | 0.99 [0.97–1.01] p-value = 0.56 | 0.91 [0.68–1.24] p-value = 0.56 | 0.98 [0.87–1.09] p-value = 0.95 | 0.97 [0.69–1.38] p-value = 0.80 | 1.07 [0.88–1.29] p-value = 0.85 | 0.96 [0.77–1.21] p-value = 0.45 | 1.09 [0.76–1.56] p-value = 0.63 | 0.94 [0.75–1.16] p-value = 0.56 |
| Embryo quality[b] | 1.03 [0.98–1.08] p-value = 0.22 | 0.99 [0.97–1.02] p-value = 0.68 | 1.07 [0.72–1.59] p-value = 0.75 | 0.99 [0.86–1.14] p-value = 0.88 | 1.12 [0.74–1.72] p-value = 0.71 | 1.06 [0.83–1.35] p-value = 0.18 | 1.14 [0.86–1.51] p-value = 0.32 | 0.98 [0.62–1.55] p-value = 0.93 | 0.87 [0.66–1.15] p-value = 0.34 |
| Pregnancy | 1.08 [1.03–1.13] _p-value = 0.003_ | 1.01 [0.98–1.03] p-value = 0.66 | 0.29 [0.18–0.46] _p-value < 0.001_ | 0.91 [0.78–1.07] p-value = 0.15 | 1.08 [0.69–1.69] p-value = 0.07 | 1.44 [1.14–1.82] _p-value < 0.001_ | 0.99 [0.71–1.39] p-value = 0.93 | 1.30 [0.77–2.20] p-value = 0.32 | 1.50 [1.01–2.24] p-value = 0.05 |
| Live birth | 1.14 [1.06–1.22] _p-value < 0.001_ | 1.01 [0.98–1.04] p-value = 0.41 | 0.21 [0.11–0.43] _p-value < 0.001_ | 0.79 [0.64–0.97] p-value = 0.28 | 0.52 [0.24–1.13] p-value = 0.10 | 2.10 [1.51–2.93] _p-value < 0.001_ | 1.04 [0.72–1.49] p-value = 0.45 | 1.77 [0.93–3.36] p-value = 0.08 | 1.86 [1.10–3.15] _p-value = 0.02_ |

Relative risk (95% CI)

| Outcomes | Weekly use of face cream | Weekly use of cleansing lotion | Weekly use of body lotion | Weekly use of perfume | Weekly use of foundation cream | Weekly use of lip and eyeliner | Weekly use of mascara | Weekly use of lipstick |
|---|---|---|---|---|---|---|---|---|
| Oocyte fertilization | 0.96 [0.85–1.09] p-value = 0.54 | 0.98 [0.88–1.09] p-value = 0.75 | 0.99 [0.87–1.12] p-value = 0.85 | 0.96 [0.86–1.08] p-value =0.49 | 0.96 [0.86–1.08] p-value = 0.53 | 0.98 [0.88–1.09] p-value = 0.72 | 0.99 [0.89–1.10] p-value = 0.85 | 0.96 [0.87–1.06] p-value = 0.61 |
| Embryo quality[b] | 1.02 [0.86–1.20] p-value = 0.86 | 1.04 [0.91–1.20] p-value = 0.55 | 0.94 [0.80–1.10] p-value = 0.43 | 0.95 [0.82–1.10] p-value = 0.49 | 1.05 [0.92–1.21] p-value = 0.47 | 1.08 [0.95–1.23] p-value = 0.24 | 1.03 [0.90–1.18] p-value = 0.69 | 0.95 [0.83–1.08] p-value = 0.97 |
| Pregnancy | 0.67 [0.55–0.81] _p-value < 0.001_ | 0.74 [0.63–0.88] _p-value < 0.001_ | 1.29 [1.06–1.57] _p-value = 0.01_ | 1.07 [0.89–1.12] p-value = 0.48 | 0.77 [0.62–0.94] _p-value = 0.01_ | 0.73 [0.59–0.90] _p-value = 0.004_ | 0.96 [0.82–1.12] p-value = 0.58 | 1.20 [1.02–1.40] _p-value = 0.01_ |
| Live birth | 0.50 [0.38–0.66] _p-value < 0.001_ | 0.60 [0.47–0.76] _p-value < 0.001_ | 1.50 [1.15–1.96] _p-value = 0.003_ | 1.33 [1.03–1.72] _p-value = 0.03_ | 0.72 [0.56–0.94] _p-value = 0.01_ | 0.81 [0.65–1.00] p-value = 0.05 | 1.09 [0.91–1.30] p-value = 0.35 | 1.44 [1.18–1.76] _p-value = 0.04_ |

Notes:
In italic bold _p_ < 0.05.
Poisson regression with robust error variance models with 188 degrees of freedom used to estimate relative risk (95% CI) for IVF outcomes in relation to women's lifestyle habits and behaviours.
[a] Self-reported level of psychological stress (including work-related stress) on a level of 1 (low) to 3 (high).
[b] Total number of grade 1 (best) and grade 2 (good) quality embryos based on the Istanbul consensus scoring system (_Balaban et al., 2011_).

**Table 4 Adjusted associations between lifestyle habit and behavior patterns and clinical outcomes among *n* = 35 women undergoing IVF.**

| Outcomes | Relative risk (95% CI) | | | |
| | PCP-use[a] | *p*-value | Healthy diet and physical activity[b] | *p*-value |
|---|---|---|---|---|
| Fertilized oocytes[c] | 0.99 [0.95–1.02] | 0.53 | 0.98 [0.88–1.08] | 0.68 |
| Embryo quality[d] | 1.01 [0.97–1.06] | 0.62 | 0.98 [0.85–1.12] | 0.72 |
| Pregnancy | 0.92 [0.87–0.98] | ***0.01*** | 1.10 [0.93–1.30] | 0.28 |
| Live birth | 0.94 [0.88–1.01] | 0.08 | 0.97 [0.78–1.22] | 0.80 |

Notes:
In italic bold, $p < 0.05$.
Poisson regression with robust error variance models with 187 degrees of freedom used to estimate relative risk (95% CI) for IVF outcomes in relation to women's lifestyle patterns adjusted for current cigarette smoking status (yes/no), BMI, age, and highest level of education attained.
[a] Varimax rotated principal component describing women's weekly use of several personal care products (face cream, face cleaning lotion, body lotion, perfume, foundation cream, lip and eyeliner, and mascara).
[b] Varimax rotated principal component describing women's weekly consumption of vegetables and fruit and their weekly frequency of exercise and duration of each workout.
[c] $n = 194$ oocytes.
[d] $n = 79$ embryos.

**Table 5 Adjusted associations between lifestyle habit and behavior patterns and intermediate outcomes among *n* = 35 women undergoing IVF.**

| *Outcomes* | Effect estimate (95% CI) | | | |
| | PCP-use[a] | *p*-value | Healthy diet and physical activity[b] | *p*-value |
|---|---|---|---|---|
| AMH | −0.12 [−0.16 to −0.08] | ***<0.001*** | 0.12 [0.002–0.23] | 0.05 |
| AFC[c] | −0.02 [−0.03 to −0.02] | ***<0.001*** | −0.01 [−0.03 to 0.02] | 0.70 |
| Endometrial thickness | −0.04 [−0.08 to 0.001] | 0.06 | −0.22 [−0.34 to −0.10] | ***<0.001*** |
| Peak estradiol | −26.7 [−52.6 to −0.82] | ***0.04*** | 254 [178–331] | ***<0.001*** |

Notes:
In italic bold, $p < 0.05$.
Linear regression models with 187 degrees of freedom used to estimate mean difference (95% CI) for AMH, endometrial thickness, and peak estradiol as outcomes in relation to women's lifestyle patterns adjusted for current cigarette smoking status (yes/no), BMI, age, and highest level of education attained.
[a] Varimax rotated principal component describing women's weekly use of several personal care products (face cream, face cleaning lotion, body lotion, perfume, foundation cream, lip and eyeliner, and mascara).
[b] Varimax rotated principal component describing women's weekly consumption of vegetables and fruit and their weekly frequency of exercise and duration of each workout.
[c] Negative binomial regression used to estimate expected difference (95% CI) in antral follicle count as the outcome in relation to women's lifestyle adjusted for current cigarette smoking status (yes/no), BMI, age, and highest level of education attained.

We retained two PCA factors from the PCA of women's lifestyle habits and behaviors (Table S2). The first PCA factor describes PCP usage in the past week, including face cream, cleansing lotion, body lotion, perfume, foundation cream, lip and eyeliners, and mascara. The other PCA factor describes women's dietary habits, comprised of weekly consumption of vegetables, fruits, and fish, and the weekly frequency and duration of exercise sessions.

Table 4 shows the confounder-adjusted associations between PCA factors summarizing women's lifestyle habits and behavior patterns and clinical IVF outcomes. Lower probabilities of pregnancy (RR: 0.92, 95% CI [0.87–0.98]) and live birth (RR: 0.94, 95% CI [0.88–1.01]) were associated with greater use of PCPs. Healthy dietary habits and greater physical activity were associated with a higher likelihood of pregnancy albeit

without statistical significance (RR: 1.10, 95% CI [0.93–1.30]). The unadjusted associations were similar to the confounder-adjusted associations (Table S3).

We also examined the confounder-adjusted associations between individual lifestyle habits and behaviors and intermediate IVF outcomes, shown in Table S4. Baseline AFC (β: −0.13 follicles, 95% CI [−0.20 to −0.06]), endometrial thickness (β: −1.18 mm, 95% CI [−1.52 to −0.85]), and peak estradiol levels (β: −677 pg/mL, 95% CI [−910 to −444]) were significantly negatively associated with a greater stress level. Similarly, regular weekly use of face cream was negatively associated with AFC and endometrial thickness (β: −0.07 follicles, 95% CI [−0.10 to −0.04] and β: −0.49 mm, 95% CI [−0.63 to −0.36], respectively). Greater use of foundation cream was also associated with lower levels of AMH (β: −0.36 ng/mL, 95% CI [−0.49 to −0.23]) and peak estradiol (β: −96.3 pg/mL, 95% CI [−189 to −3.23]). The unadjusted associations can be found in Table S5.

Table 5 describes the confounder-adjusted associations between PCA factors summarizing women's lifestyle habits and behavior patterns and intermediate IVF outcomes. AMH levels (β: −0.12 ng/mL, 95% CI [−0.16 to −0.08]), AFC (β: −0.02 follicle, 95% CI [−0.03 to −0.02]), and peak estradiol levels (β: −26.7 pg/mL, 95% CI [−52.6 to −0.82]) were significantly negatively associated with greater use of PCPs. A higher healthy diet and physical activity PCA factor score was significantly associated with greater AMH levels (β: 0.12 ng/mL, 95% CI [0.002–0.23]) and peak estradiol levels (β: 254 pg/mL, 95% CI [178–331]). The unadjusted associations between PCA factors and intermediate IVF outcomes were similar, as shown in Table S6.

## DISCUSSION

In this pilot study we report that certain lifestyle habits and behaviors were associated with IVF outcomes among Romanian women, adjusted for current smoking status, BMI, age, and education, including consumption of vegetables and fruit, stress, and use of PCPs. Unlike previously published studies (Homan, Davies & Norman, 2007; Domar et al., 2012; Hornstein, 2016), we also grouped a large number of self-reported lifestyle habits and behaviors into two independent principal component summary patterns as predictors of IVF outcomes, that represented: (1) use of PCPs (including face cream, cleansing lotion, body lotion, perfume, foundation cream, lip and eyeliners, lipstick, and mascara), potentially reflecting exposure to toxic chemical agents such as metals, phthalates, parabens, and phenols (Berger et al., 2019); and (2) healthy dietary habits and physical activity, including monthly fish consumption, weekly consumption of vegetables and fruits, and weekly exercise frequency and duration. Our results showed that greater PCP use was associated with lower probabilities of pregnancy and live birth, while healthy dietary habits and physical activity were associated with a higher likelihood of pregnancy after IVF, although without statistical significance. We also found associations between the two principal components and intermediate IVF outcomes (AFC, AMH, and peak estradiol levels), suggesting that these intermediate IVF outcomes were significantly negatively associated with greater use of PCPs, while a healthy diet and greater physical activity PCA factor score was significantly associated with greater AMH and peak estradiol levels.

### IVF pregnancy and live birth rates in Romania and other European countries

In 2016, The European IVF Monitoring Consortium (EIM) reported that the average IVF clinical pregnancy rate was 28.0% per oocyte retrieval, varying between 13.2–57.1% among countries, while the average ICSI clinical pregnancy rate was 25.0% per oocyte retrieval, varying between 18.7–42.6% among different countries (*European IVF-monitoring Consortium (EIM) for the European Society of Human Reproduction and Embryology (ESHRE) et al., 2020*). Romania initiated 3,479 IVF and ICSI cycles in 2016, with clinical pregnancy rates of 31.6% and 28.3% per oocyte retrieval, respectively, similar to those reported for other EU countries (*e.g.*, IVF pregnancy – France 21.1%, Italy 21.6, Denmark 20.9%, Germany 28.8%, Belgium 27.8%, Ireland 40.4%, Hungary 31.4%, the Netherlands 31.0%, and the UK 32.8%; ICSI pregnancy – Czech Republic 22.5%, Bulgaria 26.3%, Portugal 23.0%, Sweden 27.4%, Ireland 38.8%, Iceland 30.2%, Poland 25.8%, the Netherlands 32.9%, and the UK 33.1%). In 2016, Romania had 23.4% live deliveries per oocyte retrieval, similar to or greater than in, Poland (23.1%), Iceland (23.1%), the Netherlands (22.2%), Belgium (20.3%), Germany (19.4%), and France (18.3%), but lower than in Norway (24.2%), Sweden (24.3%), and the UK (28.5%) (*European IVF-monitoring Consortium (EIM) for the European Society of Human Reproduction and Embryology (ESHRE) et al., 2020*).

### PCP use and IVF outcomes

Europe is one of the largest markets for PCPs globally. In 2021, Germany and France consumed the greatest quantity of PCPs in Europe (based on consumption value in millions of euros), valued at nearly 13.6 and 12 billion euros, respectively, while Romania was the 11th largest consumer, with consumption valued at 1.5 billion euros. There are several European cosmetic brands which are sold and most used in every European country, including Romania (*Statista Research Department, 2022*).

Previous work suggested that PCP use may lead to greater exposure to reproductive toxicants, such as phthalates (*Begum et al., 2020*), parabens, phenols (*Berger et al., 2019*), and toxic or potentially toxic trace elements such as copper (Cu), lead (Pb), chromium (Cr), nickel (Ni), cadmium (Cd), cobalt, and arsenic used as pigments and in raw materials of cosmetics production (*e.g.*, lip cosmetics and foundations in particular) (*Ayenimo et al., 2010*; *Bocca et al., 2014*; *Gao et al., 2018*). Exposure to these agents may have adverse effects on IVF outcomes (*Bloom et al., 2010*; *Ehrlich et al., 2012*; *Björvang & Damdimopoulou, 2020*; *Begum et al., 2021*). A recent study that conducted an evaluation of metal concentration levels in different cosmetics (*e.g.*, lotions, foundations, whitening creams, lipstick, hair dyes, and sunscreens) in Pakistan, showed that sunscreens had a higher content of Ni, Pb, and Cr ($7.99 \pm 0.36$, $6.37 \pm 0.05$, and $0.43 \pm 0.01$ mg/kg, respectively) compared to other cosmetic products, lipsticks had high iron concentrations ($12.0 \pm 1.8$ mg/kg), and lotions contained high Cd concentrations ($0.26 \pm 0.02$ mg/kg) (*Arshad et al., 2020*). Another study conducted in China reported that lip cosmetics contained bioaccessible Cu, Pb, and Cr (*Gao et al., 2018*). Previous work conducted in the U.S. suggested that trace exposure to Pb, Cd, Cr, and Cu might affect maturation of oocytes

(*Bloom et al., 2010*; *Bloom, 2012*; *Ingle et al., 2017*), impact embryo quality (*Bloom et al., 2011*), and decrease the likelihood of pregnancy and live birth from IVF (*Bloom, 2012*; *Butts et al., 2021*). Consistent with previous study results, our analysis showed that a greater use of PCPs was associated with lower probabilities of pregnancy and live birth after IVF. Given the limited sample size for this pilot study, a larger investigation will be necessary to more definitively estimate the associations of PCP use with IVF outcomes among Romanian women.

## Healthy dietary habits, physical activity, and IVF outcomes

Food products in Romania are purchased mostly from supermarkets that mainly sell products imported from other European countries (*e.g.*, Spain, Italy, Greece, Poland, and Turkey). As regards the physical activity, a review of data collected from 38 European countries showed similar mean weekly durations of moderate to vigorous physical activity in Romania (599 min/week), Poland (599 min/week), and Hungary (593 min/week), which was less than in Bulgaria (675 min/week), Croatia (775 min/week), Germany (637 min/week), Greece (667 min/week), and the Netherlands (960 min/week), but greater than in Italy (212 min/week), France (259 min/week), Spain (357 min/week), the UK (259 min/week), Denmark (468 min/week), and Austria (499 min/week) (*Loyen et al., 2016*).

Previous studies indicated a potential relationship between diet and infertility treatment outcomes, although the results have been mixed (*Sanderman, Willis & Wise, 2022*). A diet rich in unsaturated fats, vegetables, fish, and whole grains was associated with a positive impact, whereas a diet rich in saturated fats and sugar had a negative impact on fertility outcomes in both women and men (*Gaskins & Chavarro, 2018*). Fish intake, likely due to its omega 3 polyunsaturated fatty acids content, was associated with a greater likelihood of blastocyst formation (*Braga et al., 2015*), and also, with a greater likelihood of live birth in women undergoing IVF (*Nassan et al., 2018*). Vitamin D3 in fatty fish may also play an important part in female reproduction (*Anagnostis, Karras & Goulis, 2013*) by influencing follicle development (*Irani & Merhi, 2014*). However, the beneficial effects of fish consumption may be overshadowed by exposure to reproductive toxicants found in seafood, including mercury (*Al-Saleh et al., 2008*; *Butts et al., 2021*) and polychlorinated biphenyls (PCBs) (*Meeker et al., 2011*; *Bloom et al., 2017*). In this pilot study, we found that monthly fish consumption was associated with higher likelihoods of good quality embryos and live birth, albeit non-significant. Greater vegetable and fruit intake was associated with better embryo quality in a study of 269 Brazilian IVF patients (*Braga et al., 2015*). However, increased exposure to pesticide residues on vegetables and fruit has also been associated with a lower probability to achieve a clinical pregnancy and live birth from IVF in a U.S. study (*Chiu et al., 2018*). Here, we found that greater consumption of fruits and vegetables was associated with higher probabilities of pregnancy and live birth (with statistical significance for fruit consumption).

Regular exercise helps to prevent energy excess and heightens sensitivity to insulin, which is beneficial for reproductive function (*Redman, 2006*). Also, regular exercise programs and changes in lifestyle leading to weight loss in overweight and obese women

have a positive impact on menstrual function, improving ovulation and subsequent fertility (*Silvestris et al., 2018*). Women's physical activity before initiating IVF was associated with higher pregnancy and live birth rates in a meta-analysis of eight studies, and also with a small increase (not statistically significant) in the implantation rate (*Rao, Zeng & Tang, 2018*). Yet, results from a population health survey comprising 3,887 women under the age of 45 years indicated an association between robust exercise (daily exercise or exercise until extreme tiredness) and subfertility, although no associations were reported with reduced intensity exercise (*Gudmundsdottir, Flanders & Augestad, 2009*). Also, in a prospective cohort study including 2,232 women, at least 4 h per week of higher intensity exercise for a year or more preceding the IVF cycle was related to greater (2.8-fold) odds of cycle cancellation, a 2-fold increase in implantation failure, and a 40% decrease in live births after the first IVF cycle compared to women who did not exercise regularly (*Morris et al., 2006*). Our analysis of individual lifestyle habits and behaviors and clinical IVF outcomes showed that routine weekly exercise was associated with a non-significant lower likelihood of live birth. However, in the PCA analysis including physical activity together with healthy dietary habits as one principal component, we found that combined healthy diet and greater physical activity was associated with a non-significant higher likelihood of pregnancy, and significantly greater AMH and peak estradiol levels. Still, given our small pilot sample, a larger investigation is needed to more definitively estimate the impact of dietary habits and physical activity on clinical and intermediate IVF outcomes among Romanian women.

### Strengths and limitations

Though a pilot study, our work has several strengths. We used a prospective study design to capture periconceptional events and thus, to ensure temporality for most IVF outcomes. However, our statistical power to detect modest associations between women's lifestyle habits and behaviors with IVF outcomes was limited by the small sample size, and thus we did not adjust for potentially chance findings from multiple statistical testing errors (*Goldberg & Silbergeld, 2011*). However, our aim was to identify plausible hypotheses for future confirmation in a larger, adequately powered investigation of lifestyle factors and IVF outcomes among Romanian women. We detected several unexpected associations, such as positive relations between past and passive tobacco smoke exposure and IVF outcomes, despite the deleterious impact of cigarette smoking on IVF outcomes (*Penzias et al., 2018*). A larger and more complex investigation incorporating a biomarker of tobacco smoke exposure among Romanian IVF patients, such as urinary cotinine, is necessary to more clearly assess the risk. Also, we did not collect detailed data on male partner lifestyle habits and behaviors, which may also impact IVF outcomes (*Firns et al., 2015*). Our study questionnaire was not previously validated and may have failed to accurately capture important lifestyle habits and behaviors, leading to exposure misclassification and possibly residual confounding. However, the study questionnaire was designed to query short and long-term habits and behaviors among Romanian women that were potentially associated with IVF outcomes. A future investigation using validated instruments will be necessary, to more accurately capture PCP use, diet, stress, and

physical activity among Romanian IVF patients. Finally, our study population was enrolled from a single IVF treatment center and so may not be representative of all Romanian IVF patients. However, we enrolled women aged 27–44 years from across the Transylvania region, representing nine of 41 Romanian counties, offering pilot data to compare to future study populations.

## CONCLUSIONS

In this pilot study we identified associations between IVF outcomes among Romanian women and certain lifestyle habits and behaviors including stress, diet and physical activity, and use of certain PCPs. We also estimated the joint effects of multiple lifestyle factors using PCA and found that PCP use and healthy dietary habits and physical activity were associated with IVF outcomes. At present, very few data are available to inform potential interventions to improve live birth rates among Romanian IVF patients. These preliminary results help to lay the groundwork for a more definitive future investigation of lifestyle habits and behaviors among Romanian women that may be of use to improve IVF success rates.

## ACKNOWLEDGEMENTS

We would like to thank the study participants, whose generous time and effort made this pilot study possible.

### Funding

The authors received no funding for this work.

### Competing Interests

The authors declare that they have no competing interests.

### Author Contributions

- Iulia A. Neamtiu conceived and designed the experiments, performed the experiments, analyzed the data, prepared figures and/or tables, authored or reviewed drafts of the article, and approved the final draft.
- Mihai Surcel conceived and designed the experiments, performed the experiments, authored or reviewed drafts of the article, and approved the final draft.
- Thoin F. Begum analyzed the data, prepared figures and/or tables, and approved the final draft.
- Eugen S. Gurzau conceived and designed the experiments, authored or reviewed drafts of the article, and approved the final draft.
- Ioana Berindan-Neagoe conceived and designed the experiments, authored or reviewed drafts of the article, and approved the final draft.
- Cornelia Braicu conceived and designed the experiments, authored or reviewed drafts of the article, and approved the final draft.

- Ioana Rotar performed the experiments, authored or reviewed drafts of the article, and approved the final draft.
- Daniel Muresan performed the experiments, authored or reviewed drafts of the article, and approved the final draft.
- Michael S. Bloom conceived and designed the experiments, authored or reviewed drafts of the article, and approved the final draft.

## Human Ethics

The following information was supplied relating to ethical approvals (*i.e.*, approving body and any reference numbers):

The "Iuliu Hatieganu" University of Medicine and Pharmacy (Cluj-Napoca, Romania) granted Ethical approval to carry out the study within its facilities.

## Data Availability

The raw data is available in the Supplemental File.

## Supplemental Information

Supplemental information for this article can be found online at http://dx.doi.org/10.7717/peerj.14189#supplemental-information.

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
