# Peer review of "Specific lifestyle factors and in vitro fertilization outcomes in Romanian women: a pilot study"

_PeerJ, doi:10.7717/peerj.14189_

## Round 0.1 · original submission · Minor Revisions

Thank you very much for the submission. In reviewing the information provided by the reviewers, I am requesting revisions before the manuscript can be considered further. Please carefully revise the manuscript according to the reviewer's comments.

Reviewer 1 ·

Basic reporting

Intro: Intro is well framed and supported with excellent highlighting of broad (global) relevance and significance of the health problem. However, while yes, this is a manuscript reporting is for a pilot study conducted in Romania. What is missing is small amount of broader context for the specifics of this pilot that help to offer up this pilot as a model for broader interest.
—Suggest providing context as to whether IVF procedures are similar in Romania as more broadly in the EU.
—Suggest Providing context as to whether the lifestyle guidance surrounding IVF is similar in Romania as is across the EU
Results:
• There are too many supplemental tables and descriptions placed between the primary tables. Supplemental Table 3 seems out of place since data for the intermediate. IVF variables hasn’t been presented in its primary form until Table 4. Moreover, supplemental Table 3 is occurring before primary Table 3 so it is an awkward construction for the reader.
—Suggest reorganizing to place the tables for the intermediate IVF variables together with bridging text to make it clear that is how the presentation is flowing. Note: This reorganization would appear to make sense given the way authors have framed the beginning paragraph of the discussion section.

• Text in the results section for Table 2 is too thin.
—Suggest revising to provide more detail so the reader can read the section before looking at the table.

• Current Supplemental Table 1 really goes hand-in-hand with Table 2
—Suggest including in the primary material (not supplemental) for the article.
Discussion:
• “Exercise….. the larger study… 4 or more hours per week….” — hours per week is missing
• Note: the OR etc. in this section of the discussion don’t match the numbers reported in the article’s abstract, Reviewer doesn’t have access to paper so maybe these are still correct
—Suggest verifying numbers provided here match the published research findings

Experimental design

no comment

Validity of the findings

Discussion:
• There is no mention of how the absolute outcome results “scale” with normal clinical practice in Romania, and EU. Is the rate of pregnancy and live birth in the expected range? It would appear the pregnancy rate is https://www.eurekalert.org/news-releases/543795 .... The live birth rate also seems reasonable based on large UK data https://www.ncbi.nlm.nih.gov/pmc/articles/PMC4934614/
—Suggest the inclusion of absolute scaling context to increase the strength of the pilot’s generalizability

—Suggest the inclusion of whether PCPs are similar across EU or specific to Romania? Same with general population lifestyle preferences for physical activity and diet.

Additional comments

no comment

Reviewer 2 ·

Basic reporting

The Authors take up the interesting and socially important topic related to infertility treatment.
This work has potential but needs a few improvements.

Introduction
1. Infertility affects both women and men, so I suggest using "Many couples" instead of "Women" in the first sentence (line 44).
2. I also suggest that you rephrase the last sentence in this section, (Lines 75-78), as it is too long. The aim of the study should be clearly stated and be the same in the main text and in the introduction.

Experimental design

Material and methods

1. The Authors should first list the methods used in the work, and then describe them in detail in turn.
2. The used questionnaire could be better described. (Who was the author of the questionnaire? How many questions does it contain?)

Validity of the findings

No comment.

Additional comments

1. I suggest changing the title, for example:
Selected lifestyle factors and the success of in vitro fertilization in Romanian women: a pilot study.

2. The abstract is included twice in the manuscript.

3. All tables should be corrected in accordance with the guidelines of the journal (e.g. font size, italics, bold, footnotes).

---

## Round 0.2 · accepted · Accept

Thank you for resubmitting the revised version of this manuscript. The manuscript has been significantly improved, and I recommend the acceptance of this manuscript.

Reviewer 1 ·

Basic reporting

No further comment

Experimental design

no further comment

Validity of the findings

no further comment

Additional comments

Manuscript was nicely revised. no further concerns/comments

Reviewer 2 ·

Basic reporting

Suggested changes were made by the Authors.

Experimental design

Suggested changes were made by the Authors.

Validity of the findings

No comment.

Additional comments

In the light changes made I suggest accepting the article for publishing.